# Food Security Challenges and Resilience during the COVID-19 Pandemic: Corner Store Communities in Washington, D.C.

**DOI:** 10.3390/nu14153028

**Published:** 2022-07-23

**Authors:** Melissa Hawkins, Maulie Clermont, Deborah Wells, Marvena Alston, Robin McClave, Anastasia Snelling

**Affiliations:** 1Department of Health Studies, College of Arts & Sciences, American University, Washington, DC 20016, USA; mc3739b@student.american.edu (M.C.); mcclave@american.edu (R.M.); stacey@american.edu (A.S.); 2DC Central Kitchen, Washington, DC 20001, USA; dwells590@gmail.com (D.W.); marvenakalston@gmail.com (M.A.)

**Keywords:** qualitative research, COVID-19, food security, food access, community health workers

## Abstract

The COVID-19 pandemic exacerbates the complexities of food inequity. As one of the social determinants of health, food insecurity significantly impacts overall health across the life course. Guided by the Getting to Equity Framework, this qualitative community-engaged participatory project examines the impact of the pandemic on food security among adults in Washington, DC. Semi-structured interviews (*n* = 79) were conducted by trained community health workers between November 2020 and December 2021 at corner stores. Data analysis was performed using thematic network analysis in NVivo. Results are grouped into four key themes: (1) impact of the pandemic on food access, including expanded services and innovative solutions to meet needs; (2) coping and asset-based strategies at the individual and community level; (3) sources of information and support, and (4) impact of the pandemic on health and well-being. The importance of lived experience research in public health is increasingly recognized as an innovative approach that offers benefits through community engagement and empowerment.

## 1. Introduction

Food insecurity is a social and economic condition with negative health consequences across the lifespan [1,2]. The United States Department of Agriculture (USDA) [3] defines food insecurity as reduced access to affordable and nutritiously adequate food. Food insecurity in childhood is associated with increased risk for cognitive problems [2], mental health disorders [4], and poorer general health [5]. In adults, food insecurity is associated with diabetes [6], hypertension [7], heart disease [8], depression [9] obesity and other chronic conditions [2]. Further, health care usage [1] and costs [10] are higher on average among adults living in food insecure households compared to those living in food secure households.

Prior to the COVID-19 pandemic, over 35 million individuals experienced food insecurity in the US; this number increased significantly due to the pandemic, the majority of whom are children and families [11]. Food insecurity impacts low-income communities and individuals with chronic conditions disproportionately and is highly correlated with economic conditions including trends in rates of unemployment [12], income [11], and food prices [13].

The Supplemental Nutrition Assistance Program (SNAP) and child nutrition programs administered by the USDA address food insecurity at the national level. Millions of children receive breakfast and lunch in their schools and during the summer through the National School Lunch Program, School Breakfast Program, and Child and Adult Care Food Program [14]. Participation in nutrition programs has been shown to help reduce food insecurity [15] and childhood obesity [16] and improve academic performance [17].

To address food insecurity and support the nutritional health of income-eligible individuals during the COVID-19 pandemic, SNAP benefits were increased and many school districts rapidly responded to provide meals to families through innovative programs [18] (e.g., meal pick up sites, meals served off site, increases in eligibility for all children). This was supported by the federal nutrition assistance program and Families First Coronavirus Response Act (H.R. 6201 Public Law: 116–127) which allowed states to apply for waivers granting flexibilities in implementing child nutrition programs, including the Special Supplemental Nutrition Program for Women, Infants, and Children (WIC). This Act also permitted states to apply for the pandemic electronic benefits transfer (P-EBT) program which aided families of children who were eligible [19]. However, some reports demonstrate that, despite the implementation of USDA waivers, 30% fewer primary and secondary students participated in the school meal programs in the first year of the pandemic [20,21]. Further, there is a lack of systematically collected data on race and ethnicity that has challenged the understanding of the impact of COVID-19 at the community level.

### COVID-19 Pandemic and Food Security

The secondary impacts of COVID-19 are evident, reinforcing inequities due to social determinants of health (SDOH) and complexities of food inequity, especially among low-resourced communities. Rising unemployment during the pandemic heightened food insecurity in the U.S. The corresponding rise in food insecurity and disrupted food accessibility contributed to additional adverse health impacts [22]. Individuals living in healthy food priority areas (limited access to affordable and nutritious food with high-density of stores selling high-calorie fast food) [23] faced additional barriers to accessing healthy and affordable food [24]. As the COVID-19 pandemic continues to challenge every sector and aspect of life, the broader downstream consequences to those who experience food insecurity are emerging.

There are ongoing studies documenting changes in food insecurity due to the COVID-19 pandemic [25]. A U.S. study [26] among 3219 respondents in Vermont found a 32% increase in household food insecurity since the start of the pandemic in March 2020, with one-third of food insecure households classified as newly food insecure. Those experiencing a job loss were at significantly higher odds of experiencing food insecurity. Of note, the researchers found significant differences in coping strategies between those individuals in newly food insecure compared with consistently insecure households. Research conducted in countries outside of the U.S. during the COVID-19 pandemic demonstrate that marginalized communities were at high risk for severe hunger and food insecurity [27,28,29,30]. However, a recent study [31] also documented increased community resilience as indicated by positive emotions (e.g., empathy), attitudes (e.g., social responsibility), and support (e.g., cooperation) to mitigate the effects of the pandemic through local empowerment.

The current study was conducted in the nation’s capital Washington, D.C. (DC). as there is an urgency to understand food access challenges that have been exacerbated during the pandemic. Further, this work builds on longstanding partnerships and on previous work in DC to promote healthy lifestyles where people live, learn, work, worship, and shop [32,33,34,35]. DC was one of the most socio-economically stratified regions prior to the pandemic. According to the 2021 U.S. Census [36], D.C.’s population is ~700,000 residents. DC is divided into eight Wards, or neighborhoods, each with approximately 75,000 residents. Social stratification in DC is high with wealthier communities primarily located in the west, and lower-income Wards concentrated to the east (Figure 1). There is a 15-year difference in life expectancy between DC’s wealthiest neighborhoods in Ward 3 (87 years) and the most marginalized communities of Ward 8 (72 years) [37]; the U.S. life expectancy in 2019 was 79 years [38]. Approximately 11% of DC’s total area is classified as a healthy food priority and these areas are concentrated in Wards 5, 7, and 8 communities with populations who primarily identify as Black/African American and experience both high rates of poverty and chronic diseases [39].

Over 25% of individuals in Wards 5, 7 and 8 live below the poverty line and have the highest rates of chronic conditions in the city [40]; several studies have demonstrated associations between food insecurity and chronic disease, including heart disease, diabetes, hypertension, and overall health status [7,41,42]. Although DC has one of the highest median incomes in the nation, 14.5 % of residents experience food insecurity. The median annual household income among white residents is 81 times greater than median Black households, demonstrating the alarming wealth disparity [40].

There are only three grocery stores in Ward 7 and Ward 8 to serve over 150,000 residents [39]. This lack of access to affordable, healthy food has exacerbated health disparities in DC. Although grocery stores are rare in these communities, corner stores are abundant. Corner stores, also referred to as convenience stores or bodegas, are defined as small-scale stores that sell a limited selection of food and other products [43]. Corner stores are more prevalent in low-income communities. Typically, corner stores have more limited physical space and food items (less fresh produce, whole grains, and low-fat dairy, etc.) compared with supermarkets or grocery stores [39]. A recent study found that 66% of corner store customers reported shopping at their nearby corner store on a daily or almost daily basis [44]. First Lady Michelle Obama’s Let’s Move! [45] health promotion campaign notes that healthy corner store programs are an effective community-based strategy to improve health and help prevent childhood obesity. In 2016, the DC Mayor established the Resilient DC initiative [46] to bolster city responses in the face of natural and/or man-made challenges. DC was selected from more than 1000 cities around the world to become part of the 100 Resilient Cities and to receive technical and financial support to develop and implement resilience strategies with input from community leaders, subject matter experts, and residents.

## 2. Materials and Methods

This qualitative study was developed in partnership with DC Central Kitchen (DCCK) Healthy Corners Program with the overarching goal of illustrating the impact of the pandemic on DC residents’ food security by interviewing residents about their experiences at community corner stores. This study focuses on low-resourced Wards within DC that experience some of the highest rates of food insecurity and diet-related chronic diseases. These communities also experienced high rates of COVID-19 cases and mortality. The objective was to understand the evolving impacts of the pandemic on DC communities’ food security to better inform programming, policy, and community action. There is an ongoing and urgent need to describe the evolving challenges to food security in order to inform action and public policy that can mitigate these challenges in DC and nationally. Specifically, we sought to explore how the challenges of the COVID-19 pandemic intersect with factors at multiple levels, including how individuals and families are leveraging community strengths.

Guided by the Getting to Equity (GTE) framework [47,48] (Figure 2), we conducted in-person interviews with adults (*n* = 79) outside DC corner store communities from November 2020–December 2021. Semi-structured interviews, which included both open-ended and closed-ended questions, were conducted by two trained DCCK ‘Store Champions’, who are community health workers (CHWs) and residents of the community. The CHWs identify as Black females, who live in Ward 8, and are 55–65 years old. The GTE model was incorporated in the development of interview questions and as a framework for the analysis to examine multi-level themes impacting food security during COVID-19.

We employed a partnership-based collaborative approach collecting both qualitative and quantitative information, guided by the principles of community-engaged and trauma-informed participatory action research [49]. Community stakeholders provided meaningful engagement and were involved in decision-making throughout the research process, from conceptualization and question development, to analysis and dissemination of study results. This project brought together health researchers, nutrition and health advocates, local corner store staff, and community members to document food security challenges in DC. An important aspect of this work is to leverage existing community resources, such as trusted community-based organizations and CHWs.

### 2.1. DC Central Kitchen Healthy Corners Program

DC Central Kitchen (DCCK) is a non-profit organization founded in 1989 with the mission to combat hunger and poverty [50]. DCCK prepares more than 10,000 healthy meals daily for food insecure residents in DC, is a farm-to-school foodservice vendor at 16 high-need schools and is a provider of meals to 55 homeless shelters and nonprofits. During the COVID-19 pandemic, between March 2020–June 2022, DCCK distributed over seven million meals to families and children, in response to rising food insecurity rates. DCCK also offers a Culinary Job Training program to equip adults facing high barriers to employment with comprehensive culinary training and professional readiness education. Over 2000 individuals have completed the Culinary Job Training program with an 87% multi-year job placement rate.

In 2011, DCCK launched the Healthy Corners Program in conjunction with DC Department of Health to address the systemic and individual barriers to healthy food access. Healthy Corners incorporates all six USDA [43] recommended strategies for creating healthier corner store food environments. These strategies include: (1) local and state partnerships to support healthier options; (2) incentives to drive program participation (e.g., new store equipment, community recognition, price discounts); (3) nutrition education opportunities for community members and store owners; (4) improvements to inventory (e.g., fresh produce, low fat yogurt, product placement); (5) technical assistance to store owners to effectively buy, stock, price, and market healthier products, and; (6) marketing to encourage healthier items (e.g., signage, shelf tags). The DCCK program provides corner stores with necessary infrastructure, such as refrigeration and shelving, before delivering fresh produce at wholesale prices. Retailers then sell produce at affordable rates according to DCCK pricing guidelines. The ‘Store Champion’ CHWs provide marketing support, offer nutrition education sessions, and cooking demonstrations to empower residents in their shopping habits. Preliminary program evaluation results of the DCCK Healthy Corners Program indicate that residents in Wards 7 and 8 are beginning to adopt the practice of purchasing fresh produce at corner stores [51].

### 2.2. Humanities Truck

The Humanities Truck was built in 2018 as a high-tech mobile lab, funded by the Henry Luce Foundation and the Andrew W. Mellon Foundation. The Humanities Truck is a fully customized truck that serves as a mobile platform for collecting, exhibiting, and expanding community residents’ voices across DC. The truck’s custom-design creates a unique space for facilitating community interviews given the capacity to function as a recording studio, workshop and maker space, and exhibit venue. The information about the project was exhibited on a poster on the magnetized exterior wall of the truck.

### 2.3. Data Collection

The project included four phases: (1) collaboration and planning, including co-creation and refinement of the semi-structured interview questions; (2) data collection; (3) data analysis, and (4) community feedback and dissemination. We aimed to build on existing evidence on food insecurity in DC, with a focus on food access experiences during COVID-19. In addition to conducting interviews outside the corner stores, we also hosted cooking demonstrations with the support of DCCK and the Humanities Truck. Between November 2020–December 2021, we brought the Humanities Truck to twenty-one corner stores in Wards 5, 7 and 8 and spoke with residents (*n* = 79) about their experiences. Study procedures were approved by the university’s Institutional Review Board in October 2020 (IRB-2021-166). Participants provided both written and verbal informed consent.

The qualitative research method is a systematic approach to describe individual experiences within the context of their environment to provide meaning. DCCK and the research team collaboratively developed the semi-structured interview questions based on the published literature and previous data collection efforts with the community [52]. The questions were developed to provide an opportunity to describe challenges and opportunities in relation to the COVID-19 pandemic. There were also questions to explore opinions on the public health measures implemented locally, including sources of credible information, strategies used to manage and reduce the risk of COVID-19 infection, and sources of supports (Table 1). The interview questions were pre-tested in October 2020 to practice probing techniques, ensure active listening, and ensure questions were neutral and clear [53]. The questions were revised to include questions on hopes for the future, either personally, for DC, or the country, and lessons learned from the pandemic.

Interviews were conducted outdoors, with all parties wearing masks. Guidelines and best practices to limit the spread of COVID-19 were provided to participants. Data collection was ideal during this time, as individuals were experiencing the social and economic ramifications of COVID-19 and were well positioned to speak about their experiences. This timeframe captured participants’ lived experiences during the peak of the pandemic in 2020, through the duration of the strictest measures enforced in DC, school closures, prior to and after vaccine availability, and the lifting of the mask mandates.

The CHW interviewers had previous experience conducting interviews with DC community members and were trained to ensure the questions were consistent. Training included understanding the purpose and scope of the study, techniques for approaching and inviting participants, establishing trust and rapport, and probing questions during the interview. The same CHWs conducted the semi-structured interviews from November 2020 to December 2021. All participants were offered a variety of items for their time (mask with DC flag, granola bars, water, socks, gloves, coloring book and crayons, small stuffed animals, stickers).

The corner store locations were selected based on the following inclusion criteria: (1) store owner buy-in, as assessed by their engagement and dedication to the success of the DC Central Kitchen Healthy Corners Program, (2) varying geographic locations with high rates of food insecurity, and (3) good standing with USDA Food and Nutrition Service. All adults who approached the Humanities Truck were invited to participate and were provided with an informed consent form, including COVID-19 safety procedures. All interviews were audio recorded using a recorder and smartphone application. Participants were asked to provide some demographic information (first name, age, place of residence). The interview time was flexible to accommodate participants’ preferences and circumstances; interview time was 5 min, on average (ranging from 2 min to 20 min in length). The interviewers probed for clarification and elaboration. Participants were also invited to take a picture in front of the Humanities Truck at the completion of the interview.

### 2.4. Data Analysis

The audio-recorded interviews were transcribed verbatim, cross-checked for accuracy, then uploaded to Excel and exported to NVivo [54] for coding and analysis. NVivo (Version 13, QSR International Pty Ltd. NVivo (Released in March 2020). 2020. Available online: https://www.qsrinternational.com/nvivo-qualitative-data-analysis-software/home accessed 14 June 2022 )is a qualitative software that enables researchers to code data, assess interrater reliability and analyze data for key theme identification. The transcripts were coded to identify major themes [55]. Neither codes nor the codebook were developed a priori; the codes were derived from the data, not researcher derived. Coding protocols were followed by two coders to reduce coding inconsistencies. The researchers met weekly to discuss codes and the findings of this study. Inter-rater reliability was measured within NVivo using a proportion of agreement calculation, with a requirement of a minimum of 90% agreement to ensure coding reliability [56]. The codes were then applied to the remaining transcripts. Coding matrices were used to sort the dominant codes into categories and subcategories, identify relationships between codes, and support content analysis for key theme identification. Two coders led codebook development with feedback from the research team members and DCCK. Following blinded coding, coders identified themes that were discussed and modified [57]. Themes were shared with DCCK partners for feedback.

Descriptive statistics were calculated to summarize demographic characteristics. The analysis included reviewing the transcripts multiple times, identifying meaningful quotes, categorizing the quotes within the key concepts, organizing overarching themes, and examining relationships between themes. There were a variety of approaches utilized to enhance the validity and reliability of the analysis. For example, accuracy in both form and context was verified with blinded comparisons between coders; results were examined throughout the analysis, given the long study period; meaningful quotes were identified for each theme. The project details, pictures, and key findings were compiled in a public web exhibit (accessed on 14 June 2022)) that was first shared with participants.

## 3. Results

Community members were interviewed in locations with high risk for food insecurity during the pandemic. The participant characteristics are detailed in Table 2.

The key results related to food access are summarized in Table 3 based on the GTE framework quadrants, with key themes identified and exemplary quotes. Participants described strategies to optimize food access during the pandemic. Strategies to increase access and healthy food options included home delivery, meal pickup at a variety of convenient locations, and timely information on meal site locations. Local and federal COVID-19 specific initiatives were identified as a source of financial and emotional support. Further, participants noted family, friends, and places of worship as a source of emotional strength and support. Many participants specifically mentioned the DC Mayor and local councilmembers as source of trusted information for guidance and decision making. Reducing deterrents related to providing accessible, available and affordable food items, including fresh produce. Participants commonly described challenges in accessing services such as the DC unemployment service, access to fresh produce in corner stores and WIC. Community capacity was strengthened during the pandemic through community organizations and partnerships. Most participants identified accessing “the news” online as their primary source of trusted information, followed by family, friends and healthcare providers. Many residents mentioned resilience and the strength in community bonds. Community partnerships increased access for DC’s most vulnerable Wards to meet residents’ needs.

Figure 3 illustrates the one-word participants used to describe their COVID-19 experience. A word cloud is a visualization tool to summarize text in which the more frequently used words are noted by prominence in the figure.

## 4. Discussion

We conducted an in-depth qualitative exploration of lived experiences of DC adults during the COVID-19 pandemic in one of the most impacted areas of the U.S. The interviews provided the opportunity to expand understanding of the experience of living in a healthy food priority area and how residents use the corner store to improve food access.

Participants noted increased needs to support themselves, their families, and to manage the impact of the pandemic, while also noting the community strengths as building resilience. Emotional resilience and community resilience was challenged across every global sector during the COVID-19 pandemic. The concept of community resilience generally refers to the ability of connected groups of people to respond to cope with and recover from adverse challenges to their community [58]. Community resilience is associated with expanded local capacity [59], social support and gratitude [60,61], and increased access to resources [62], all of which were illustrated by participant responses in the current study. Patel et al. [63] offer a combination of nine core protective elements of community resilience, which vary in complexity, to measure: local knowledge, community networks, communication, health, governance/leadership, resources, economic investment, preparedness, and mental outlook. The pre-existing health of the community is also important for understanding and addressing community resilience.

The association between income and food availability is well-established [1,10,11,12]. Location is another important factor in food security; individuals who live in areas that combine high concentrations of poverty with low access to grocery stores, such as Wards 7 and 8 in DC, are more likely to experience food insecurity. Food insecurity is complex and multifactorial; in health food priority areas, community driven initiatives to increase healthy options in corner stores may help to fill food access gaps. A cross-cutting recommendation at the policy level includes the imperative for local policymakers to advocate for feasible approaches to incentivize grocery stores to locate in low-income communities.

Further, gaining a deeper understanding of pandemic challenges in under-resourced communities from community members may inform food access programming at the community-level to better serve communities in the future [64]. Historically, corner store retailers have faced barriers to selling healthy food for reasons including a lack of refrigeration space, knowledge of produce handling practices, and an inability to meet produce wholesalers’ minimum orders [39,43]. DCCK is well positioned to fill the gap in resources, knowledge, and trust to increase healthful food items for sale [44,51]. Community-driven initiatives and programs with cross-sector collaboration to increase food access, including the DCCK Healthy Corners Program, may reduce barriers to food access while simultaneously supporting the local economy, and have the potential to significantly reduce food insecurity. They can also serve as a model for other community organizations in other similar urban areas. As exemplified during the pandemic response in DC, cross-sector collaborations, across health care, social service, non-profit and governments sectors, comprehensively addressed many of these challenges.

This study also emphasizes that CHWs are poised to build and sustain trusting relationships between individuals, communities, and health systems. It has been previously shown that individuals benefit from relationships with people who have similar lived experiences and are members of their community [65]. CHWs are also uniquely qualified to address food security because they often live in the communities they serve and understand the social and cultural context. Thus, CHWs are well positioned to play a pivotal role in building trust and addressing SDOH faced by communities that experience food insecurity, especially in low-income countries with vulnerable health systems [66]. The CHW field is gaining attention as an effective solution to many of the challenges presented by SDOH, including food insecurity [67,68]. CHWs have been active in the US and worldwide for many decades and are increasingly recognized as an essential strategy for improving health and food security in communities by linking health care and community resources. The notion of integrating non-clinical workers into healthcare systems represents a cultural shift in the U.S.; however, the enactment of the 2010 Patient Protection and Affordable Care Act (ACA) and the establishment of the Triple Aim have formally recognized CHWs as valuable partners in health care. In April 2022, the American Rescue Plan provided over $225 million to launch the Community Health Worker Training Program.

During the pandemic, community-based health care organizations supported measures to reduce food insecurity which can be continued after the pandemic by leveraging structural assets, addressing systemic barriers, and connecting across sectors. The school environment also plays an important role in food security, particularly since the passage of the Healthy, Hunger-Free Kids Act in 2010, which improved the quality of meals provided by the National School Lunch Program and has been shown to attenuate the trajectory of childhood obesity, particularly among those students who are at higher risk [16]. Universal free school lunch supported families during the pandemic and may help improve food insecurity rates [69]. Furthermore, streamlining eligibility information to help cross enroll individuals for other benefits for which they qualify would contribute to program efficiency, improve access, and reduce administrative costs, reducing deterrents to healthy nutrition behaviors. Some food access programs currently use another eligibility determination to eliminate a duplicative process, but the determinations are state-specific and are challenging to navigate across programs. Healthcare providers and community-based programs can also identify SDOH and address food insecurity through simple interventions. For example, conducting a brief screening for food insecurity and providing resources may effectively reduce potential long-term impacts on health outcomes associated with the duration of household food insecurity, particularly in childhood [70], and health care costs associated with food insecurity [1]. The widely used and validated Hunger Vital Sign [71] is a two-question food security screening tool that can quickly determine risk for food insecurity in community settings. Integrating this tool in community programs would help to connect individuals experiencing food insecurity to available resources.

### 4.1. Strengths and Limitations

This study supported a community organization–academic partnership working towards health equity in DC. This works elevates community voices to illuminate the impact imposed by COVID-19 in responding to local food insecurity needs, and to strengthen social connections to the community. DC residents have increasingly called for expanded access to healthy, local foods and social resources to address the city’s ‘grocery gap.’ Study strengths include the timeliness and contribution to the limited literature on strategies to address food insecurity in low-resource urban communities during the COVID-19 pandemic. Communities are dynamic places and projects in natural environments are prone to many challenges, notwithstanding the COVID-19 pandemic. The project team was flexible and creative with data collection approaches. CHWs played a critical role in facilitating participant engagement. It is important to note that there was enthusiasm and a strong willingness to participate in this project. Residents were eager to share their stories and lived experiences.

Study limitations include the convenience sample which contributes to a lack of generalizability beyond the DC community. The in-person interviews may have amplified participants’ desire to express themselves in a socially desirable manner which may have limited participants from freely expressing negative views. Furthermore, in all qualitative research it is important to acknowledge the role of the interviewers in the research process. That is, the characteristics and experiences of the interviewers could influence both the information shared and the findings. Food insecurity was not assessed with a validated instrument.

### 4.2. Implications for Future Research

Food insecurity is one of the leading health and nutrition issues in the US. The short-term impact of the pandemic is evident, but the long-term consequences including how we will reshape and reimagine health policy priorities in this country are evolving. Although food insecurity is more prevalent in low-income communities, the pandemic and resulting economic disruptions have exacerbated the complexities of food inequity. As illustrated through the GTE framework, it is imperative to support community-driven food security strategies to promote food justice. Local and nonprofit organizations within the DC region expanded their programming and services to offer innovative solutions to meet the increasing needs of DC residents, including increased access through corner stores, meal delivery sites, and home delivery. This project has helped to center local DC food narratives on resilience and community empowerment. Leveraging external resources, partnering with health care systems and community organizations, and investing in CHWs could lead to a more resilient food system that can reduce food insecurity and improve health and well-being. Additional research is needed to understand food access barriers at multiple levels, especially among those that may be newly food insecure since COVID-19.

## Figures and Tables

**Figure 1 nutrients-14-03028-f001:**
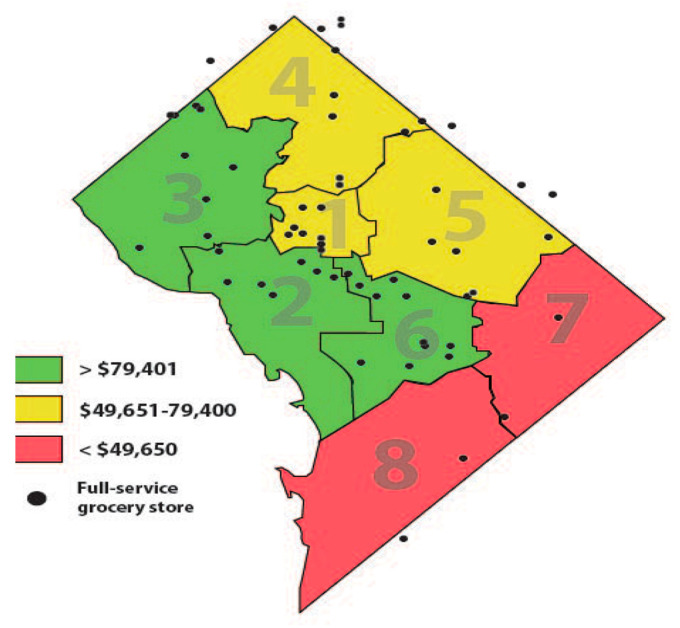
Washington DC median annual household incomes by eight Wards and grocery store locations.

**Figure 2 nutrients-14-03028-f002:**
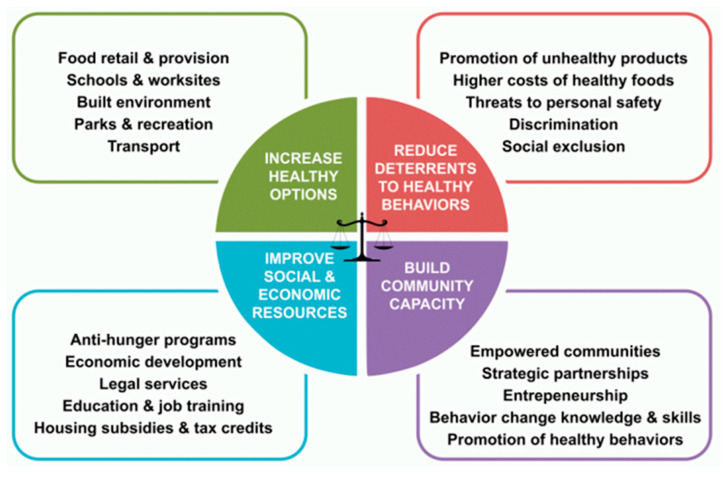
Getting to Equity Framework.

**Figure 3 nutrients-14-03028-f003:**
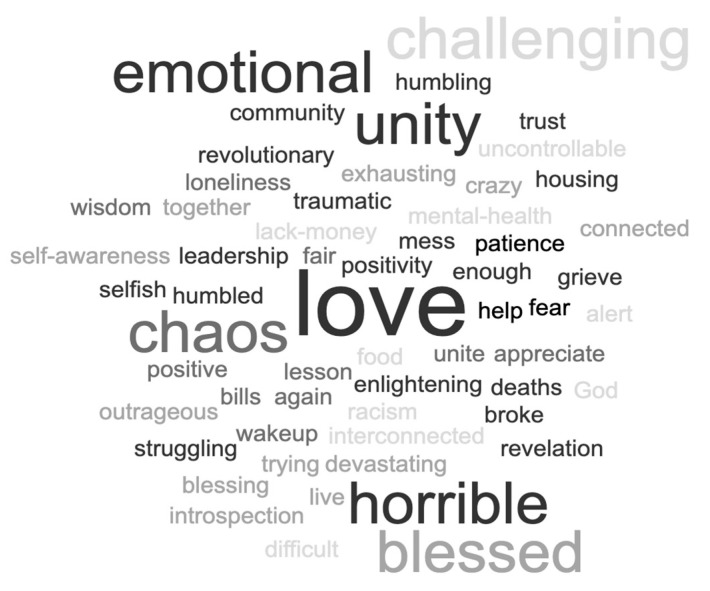
Describe the COVID-19 Pandemic in One Word (Word Cloud).

**Table 1 nutrients-14-03028-t001:** Semi-structured interview questions.

Category	Questions
Demographic	First name, age, neighborhood
Challenges	What has been biggest challenge to you/family during the pandemic? Is there anything that has been good for you/family during the pandemic?
Food Access	How are you navigating food access for you/family? How well do your current grocery store options fit your needs? How often do you shop at corner stores?
Information/Resources	Where or from who to do you get information you can trust?
Feelings/expectations	What do you expect from your family/community/politicians/institutions during this Covid pandemic?
Fears/hopes/lessons	What is your biggest fear about Covid going forward? What is your biggest hope for you/DC/US/world after the pandemic? What gives you strength right now? What is something you hope will be different for you/DC/US/world after pandemic?
Public Health	How would you describe public health? What does public health mean to you?
Specific: DC Programs, Vaccination, Schools	What do you think of the way DC has managed Covid response?
One word	What is one word to describe this time for you?
Final remarks	Is there anything that we have not covered? Would you like to share anything else?

**Table 2 nutrients-14-03028-t002:** Participant Characteristics (*n* = 79).

	*n* (%)
**Gender**	
male	31 (39%)
female	48 (61%)
**Age**	Mean = 42.4 (20–73 years)
**Corner Store/Grocery Location**	
(21 truck visits)	
Ward 5	6 (25%)
Ward 7	6 (25%)
Ward 8	9 (43%)

**Table 3 nutrients-14-03028-t003:** Key themes and participant quotes by GTE quadrant.

GTE Framework Quadrant	Key Themes	Exemplary Quotes
Increase Healthy Options	Multiple site and pick-up options: Grab-and-go meals, bus stop pick-up, multiple sites improved meal accessibility Parents described wanting healthy and fresh food and snacks for their children Challenges accessing services during COVID-19 and impact on health and well-being	“DC told us where to get the food. Transportation was free. Food was free. I was eating healthier through this. I didn’t expect that.” “The biggest challenge is high price of food and getting around to get it. There has been a lot of help in the neighborhood with people giving out food. They’ve been handing out vegetables and produce and that’s been a big help.” “I get food stamps so that’s a big help. They’ve increased them and I wish that it could go longer before it goes back. It’s helped my pantry tremendously. To be able to get organic fresh produce is so good.”
Reduce Deterrents to Healthy Behaviors	Trusted and credible sources of information, resources, and knowledge of program options Increase availability, affordability, and accessibility to fresh produce.	“At my grocery store you can tell they know what neighborhood they are in. The meat is spoiled, I have brought management attention to multiple expired items on entire shelves, molded cucumbers. You expect them to treat you like they treat you.” “It’s a class thing. They don’t give us the freshest produce. They give us soft apples…or food that expires in a matter of days. You get tired of traveling outside of your community going all the way uptown to get the freshest produce. It’s just sad.” “I have a problem with proliferation of liquor stores in the middle of a food desert. There are 7 liquor stores in one block, but no grocery stores. You have to throw a search party to find fruit in this neighborhood.”
Improve Social and Economic Resources	Pandemic-electronic benefit transfer (P-EBT) were appreciated Improved access to health care (tests, testing sites, vaccines, health care) and food through expanded programs Nutrition program waiver flexibilities	“The majority of people who are food insecure in DC are working class people, not homeless. The system shames you if you are poor. I’m hoping we see a change in our value system.” “DC and the Mayor have been doing an amazing job, getting support whether you are resident or not from providing food to basic sanitation items. They made sure everyone had a meal or place to pick up healthy food. I helped pass out too, so I saw firsthand what a difference getting help with food made. We made it ok for people to be comfortable asking for help.” “We dabbled in universal access to health care in this pandemic and we saw it’s important and effective.”
Build Community Capacity	Increased community cohesiveness and resilience Increased knowledge of public health, COVID-19 safety, and hygiene practices Positive coping strategies during the COVID-19 pandemic	“It’s been a reminder to me how interconnected everything is.” “I can’t rely on grocery stores to have food, so had to rely on kindness of neighbors, kindness of government. We saw the best and worst in society coming out during the pandemic.” “The corner stores are selling more healthy food now. I can get grapes for their snacks now. I love that they did that now. That tells me they do care about the community and our children.

## Data Availability

Data are available upon request. The data presented in this study are available by request from the corresponding author.

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
