# Peer review of "Food Security Challenges and Resilience during the COVID-19 Pandemic: Corner Store Communities in Washington, D.C."

_nutrients, 2022, doi:10.3390/nu14153028_

Round 1
Reviewer 1 Report
VIWER
1 – Pg 3, lines 89 to 91: Add information about social strata.
2- In a quick search we found studies in several countries that evaluated the impact of the Covid-19 pandemic on food insecurity. Add studies in developed, developing and underdeveloped countries.
3 – In Figure 1, add information about the periodicity of the income (annual, monthly, etc.).
4 – Pg 3, lines 103 to 108: Are the aforementioned chronic diseases associated with food insecurity? Add information about the country’s life expectancy and between life expectancy between DC’s wealthiest neighborhoods in Ward 3 and the most marginalized communities of Ward 8.
5 – Pg 3, lines 103 to 109: Briefly describe what is available for sale in grocery stores and corner stores. The authors suggest that corner stores only sell “unhealthy” foods.
6 – Pg 4, lines 111 to122: The excerpt has 2 objectives for the study. Make it clear which of these is the general objective of the study.
7 – Pg 4, lines 123 to 130: Briefly describe the characteristics of the researchers “trained DCCK ‘Store Champions’, who are community health workers (CHWs) and residents of the community.
8 – Pg 4, lines 161 to 170: Briefly describe all six USDA [2016] recommended strategies for creating healthier corner store food environments and The 'Store Champion' CHWs provide marketing support, offer nutrition education sessions, and cooking demonstrations to empower residents in their shopping habits.
9 – Pg 4, lines 161 to 170: Preliminary results of your study?
10 – Pg 5, line 192 to 193: Add the Research Ethics Committee approval number.
11 – Pg 5, lines 202 to 204: What exactly does “the interview questions were piloted informally” mean? Is this procedure adequate to guarantee face and content validity with the CHWs? Add references that corroborate the statements.
12 – Pg 5, line 204: The question is about the future of Covid-19? Describe it better.
13 – Pg 7, lines 219 to 230: Does the phrase: “varying geographic locations based on high rates of food insecurity”, mean that the Wards 5, 7, and 8 communities were selected?
14 – Add the reference or company that owns the NVivo software.
15 – Make it clear whether the categories of analysis were created a priori or a posteriori.
16 – In terms of reliability, were the results of each researcher measured blindly?
17 – Did NVivo create the analysis categories?
18 – Pg 12, lines 306 to 307: Better explain the meaning of resilience in the sentence: “while also noting the community strengths as building resilience”.
19 – Pg 12, lines 309 to 312: Add a reference that corroborates the statement.
20 – Pg 12, lines 312 to 314: Add a reference that corroborates the statement.
21 – Pg 12, lines 315 to 320: In Figure 3, what does Love mean? Or even, what is your analysis of the perception of the speech of the research participants about the Covid-19 Pandemic? Do you believe that the participants perceive social inequities? What kind of proposal can be leveraged from the participants' experiences. Is there clarity on the role and importance of public policy efficiencies to reduce food insecurity?
22 – Pg 12, lines 315 to 320: What part of the participants’ responses led the authors to the following part of the discussion: “We also illustrated the importance of CHWs in bridging between individuals, communities, and formal systems to build and sustain trusting relationships. CHWs possess deep knowledge of community strengths and risks, can provide information on navigating resources in culturally competent ways, and mobilizing advocacy efforts for community needs”.
23 - Add the Recommendations to references or previous studies in the country or in other countries that support the proposals pointed out by the authors.
24 – Discussions do not support Recommendations
Author Response
Thank you for the detailed comments and thoughtful suggestions to revise the manuscript. Please see the attachment with point-by-point responses to address each comment.

Reviewer 2 Report
The manuscript deals with an interesting and actual topic, however, several issues might be concerned:
- there are many phrases that are unfamiliar to non-US readers, therefore a clear explanation should be provided: corner store, ward, etc.
- the design of the research should be explained more - why Washington DC was selected, this is not supported
- in table2 what does 'picture' means? In case this refers to whether the interviewee allowed to take a photo - why is it relevant for the research?
- table3 in its present form is too long and hard to follow, it might be moved to the appendix and only a summary of NVivo results should be provided
- the recommendations should be linked more to the results of the research
Author Response

(The authors gave the same response as above.)

Round 2
Reviewer 1 Report
The authors responded adequately to the questions and suggestions of the reviewers, making the text clear and with accurate information. Therefore, I am in favor of publication.
Author Response
Thank you for your feedback and for taking the time to review our manuscript.
Reviewer 2 Report
Most of my concerns were addressed during the review process.
However, I still do not understand the relevance of whether the interviewee allowed to take a photo and therefore why it is an important characteristic to be presented in Table 2.
Author Response
Apologies, I uploaded the table addressing the 1st round of reviewer comments and our response to all of Reviewer #2 comments. For this review, we have deleted the information about a picture and revised Table 2. The point is well-taken that this is not relevant to the research findings presented. The attached file includes the revised Table 2. Thank you.
